# Structural and Functional Analysis of the MADS-Box Genes Reveals Their Functions in Cold Stress Responses and Flower Development in Tea Plant (*Camellia sinensis*)

**DOI:** 10.3390/plants12162929

**Published:** 2023-08-13

**Authors:** Juan Hu, Qianqian Chen, Atif Idrees, Wanjun Bi, Zhongxiong Lai, Yun Sun

**Affiliations:** 1Key Laboratory of Tea Science in Fujian Province, College of Horticulture, Fujian Agriculture and Forestry University, Fuzhou 350002, China; vicky232022@163.com (J.H.); biwanjun9701@126.com (W.B.); 2College of Life Sciences, Fujian Agriculture and Forestry University, Fuzhou 350002, China; 3190515007@fafu.edu.cn; 3Guizhou Provincial Key Laboratory for Agricultural Pest Management of the Mountainous Region, Scientific Observing and Experimental Station of Crop Pest in Guiyang, Ministry of Agriculture and Rural Affairs, Institute of Entomology, Guizhou University, Guiyang 550025, China; atif_entomologist@yahoo.com; 4Institute of Horticultural Biotechnology, Fujian Agriculture and Forestry University, Fuzhou 350002, China

**Keywords:** MADS-box, tea plant, cold stress, floral ABCE model, transcription factor

## Abstract

MADS-box genes comprise a large family of transcription factors that play crucial roles in all aspects of plant growth and development. However, no detailed information on the evolutionary relationship and functional characterization of MADS-box genes is currently available for some representative lineages, such as the Camellia plant. In this study, 136 MADS-box genes were detected from a reference genome of the tea plant (*Camellia sinensis*) by employing a 569 bp HMM (Hidden Markov Model) developed using nucleotide sequencing including 73 type I and 63 type II genes. An additional twenty-seven genes were identified, with five MIKC-type genes. Truncated and/or inaccurate gene models were manually verified and curated to improve their functional characterization. Subsequently, phylogenetic relationships, chromosome locations, conserved motifs, gene structures, and gene expression profiles were systematically investigated. Tea plant MIKC genes were divided into all 14 major eudicot subfamilies, and no gene was found in Mβ. The expansion of MADS-box genes in the tea plant was mainly contributed by WGD/fragment and tandem duplications. The expression profiles of tea plant MADS-box genes in different tissues and seasons were analyzed, revealing widespread evolutionary conservation and genetic redundancy. The expression profiles linked to cold stress treatments suggested the wide involvement of MADS-box genes from the tea plant in response to low temperatures. Moreover, a floral ‘ABCE’ model was proposed in the tea plant and proved to be both conserved and ancient. Our analyses offer a detailed overview of MADS-box genes in the tea plant, allowing us to hypothesize the potential functions of unknown genes and providing a foundation for further functional characterizations.

## 1. Introduction

The tea plant (*Camellia sinensis* (L.) O. Kuntze) belongs to the genus Camellia (Theaceae), which comprises more than 250 species [1,2]. The tea plant is a commercially important crop the leaves of which are used to produce tea for the world’s three most widely consumed nonalcoholic beverages. Due to the highly repetitive, heterozygous, and relatively large (approximately 3G) genome of the tea plant, it is challenging to perform genome assembly on this crop. In recent years, with the revolution of next-generation sequencing technology and the efforts of researchers, the tea plant genome has developed from an initial draft genome (contig level) to the current reference genome (chromosome level) [3,4,5,6]. The increasing public availability of high-quality genome assembly and transcriptomes represent a great opportunity for elucidating the molecular basis underlying various biological processes and further accelerating tea plant research and breeding.

MADS-box is one of the most important transcription factor families and is widely distributed throughout plants. The initials of the four earliest discovered MADS-box genes constitute the most characteristic domain (MADS-domain) of this gene family, including *MINICHROMOSOME MAINTENANCE1* from *Saccharomyces cerevisiae* [7], *AGAMOUS* from *Arabidopsis thaliana* [8], *DEFICIENS* from *Antirrhinum majus* [9], and *SERUM RESPONSE FACTOR* from *Homo sapiens* [10]. Prior to the divergence of the plant and animal kingdoms, MADS-box genes were differentiated into two phylogenetic groups: type I (SRF-like or M-type) and type II (called MEF2-like or MIKC-type) [11,12]. Plant-specific MIKC-type genes encode a class of proteins comprising four characteristic domains (from N- to C-terminal): the MADS (M) domain, the intervening (I) domain, the keratin-like (K) domain, and the C-terminal (C) domain [13]. Type II MADS-box genes are long and contain five to eight exons, while type I genes have a relatively simple structure with no K domain and usually contain only one or two exons [14]. Type II genes are further divided into MIKC^C^-type and MIKC*-type genes (also named Mδ in some studies) due to the variable intervening (I) domain [15]. MIKC^C^ proteins can be further subdivided into 14 distinct subfamilies based on their phylogenetic relationships [16,17]. Type I genes can be further divided into three subfamilies: Mα, Mβ, and Mγ [12,18]. 

MADS-box genes control all major aspects of life for land plants. The most famous MADS-box genes include four different classes of floral homeotic genes (A, B, C, E), which spell out the classic “ABCE” model of floral organ identity [19,20,21]. In *Arabidopsis*, Class A genes (*APETALA1* [*AP1*] and *APETALA2* [*AP2*]) alone determine sepal identity in whorl 1 [22,23]. Class A genes overlap with Class B genes *APETALA3* (*AP3*) and *PISTILLATA* (*PI*) to control petal development in whorl 2 [24,25]. Class B and Class C *AGAMOUS* (*AG*) genes together determine petals in whorl 3, and Class C genes alone specify carpel identity in whorl 4 [26]. Class E genes (*SEPALLATA1*, *2*, *3*, 4 [*SEP1*/2/3/4; formerly named *AGL2*/4/9/3]) are involved in the specification of all four whorls of floral organs by redundantly interacting with genes of other classes to form heterotetramer protein complexes [27,28]. Generally speaking, floral organs in each of the four whorls are determined by combinations of MADS-box transcription factors with different homeotic functions, such as A-, B-, C-, and E-functions, to form homodimers and heterodimers and tetramers. With the exception of *AP2*, all ABCE homologous genes are MIKC^C^-type genes. The ABCE model is very versatile for flowering plants; even the differentiated zygomorphic orchid flower can be primarily explained using a modified ABCE model [29,30].

In *Arabidopsis,* MADS-box transcription factors are also involved in regulating leaf morphogenesis; root, seed, and embryo development; and fruit ripening [31,32]. Several MIKC^C^-type genes, such as *SHORT VEGETATIVE PHASE* (*SVP*)/*AGAMOUS-like* 24 (*AGL24*), *SUPPRESSOR OF OVEREXPRESSION OF CONSTANS1* (*SOC1*) and *FLOWERING LOCUS C* (*FLC*), are of great importance because they participate in flowering transition and flowering time by integrating various environmental signals [33,34]. Moreover, many studies on MADS-box genes of different subclades have revealed the functions of some plants in abiotic stress tolerance in different developmental processes [35]. For example, AP1, AP3, and AG homologs from tomatoes were shown to be induced by low temperatures and associated with the formation of abnormal flowers [36]. The rice *AGL12* ortholog, *OsMADS26*, acts as a regulator of stress-related responses such as drought and pathogen infections [37]. *LsMADS55 (AP1-like)* is responsible for heat-induced bolting and flowering in lettuce [38]. Several other MADS-box genes are implicated in plastic developmental processes (including vernalization, seed germination, seasonal growth cessation, and winter dormancy process) in response to seasonal changes caused by environmental conditions. For horticultural plants, these plant developmental processes and stress responses are strongly related to agronomic traits. 

The prevalence and functional diversity of members of the MADS-box family have been comprehensively investigated in many angiosperms, including model plants *Arabidopsis*, *Oryza sativa*, and *Brachypodium disachyon*, as well as some commercially important crops (*Vitis vinifera*, *Triticum aestivum*, *Musa acuminata*, etc.) [39,40,41]. Although an overview of MADS-box genes in tea plants is available [42], there remains a lack of in-depth genome-wide identification and functional characterization for tea plant MADS-box genes in a well-resolved genome. In the present study, we identified 136 MADS-box gene members from the tea plant genome and transcriptome assembly by integrating several bioinformatics methods. Phylogenetic analysis, gene structures, conserved motifs, chromosomal locations, and syntenic relationships were also analyzed. Expression patterns of MADS-box genes linked to cold stress in different tissues during different seasons were further surveyed. Our results provide a detailed overview of the tea plant MADS-box gene family, which could facilitate future functional analyses of *CsMADS* genes in different biological processes. 

## 2. Results 

### 2.1. Identification of the MADS-Box Gene Family Based on Tea-Specific Nucleic Acid HMM and the Curation of Gene Models

A raw dataset of 132 non-redundant MADS-box genes was obtained through HMMER search employing downloaded HMM profiles as a query to search proteins containing the MADS-domain or K-box domain in the tea protein database (Appendix A). In this dataset, forty-four genes were encoded for both the MADS-domain and K-box domain, seventy-nine genes for only the MADS-domain, and eight genes for only the K-box domain. In order to identify all genes that were missed in published annotations and lost their MADS-box domains or a large part of their sequences due to divergent evolution, a tea-specific nucleic acid HMM with a length of 569 bp was constructed based on the predicted mRNA nucleotide sequences of a well-aligned MADS-domain and K-box domain using hmmbuild from the HMMER v3 suite (Appendix A). The resulting HMM was adopted to probe the tea plant genome [3]. After removing sites with an e-value above 0.5, 169 genomic regions with candidate MADS domains were retained for subsequent manual verification and functional annotation. In total, 124 of 132 predicted proteins overlapped with candidate genomic regions, and the rest were excluded from posterior analyses, as it was further confirmed that only the k-box gene was preserved in these genes via screening extended genomic regions (Appendix A). For each nucleotide domain without counterparts of known genes, we performed repeated predictions with the Genewise software using genes of closely related species (grape or kiwifruit) or assembled transcripts of tea-reproductive organs. Of these, the proteins determined for twenty-seven genomic regions seemed to be functional, including five type II genes (three *AP1/FUL*-like genes, one *AGL15*, and one *FLC* gene) and twenty-two type I genes (Appendix A). 

In addition to mining novel gene loci, gene models of known genes were also evaluated and manually curated. A total of twenty-four known genes were truncated, among which sixteen genes only contained the MADS domain, and eight genes only contained the K-box domain. In four cases (*CSS0040885.1*, *CSS0001766.1*, *CSS0012111.1*, and *CSS0041561.1*), gene models were reconstructed based on transcriptome assembly. Four genes (*CSS0013866.1*, *CSS0009886.1*, *CSS0016920.1*, and *CSS0006612.1*) with only partial sequences were merged with nearby truncated genes (*CSS0033417.1*, *CSS0000825.1*, *CSS0032023.1*, and *CSS0043640.1*, respectively) to form a complete gene. Gene models of the 12 truncated genes could not be reconstructed, indicating that their functional MADS-box or K-box domains may have been lost during evolution (Appendix A). In two cases, candidates *CSS0032233.1* and *CSS0021126.1* allocated to non-chromosomal locations (Contig203 and Contig399) were discarded because their sequences were identical to those of *CSS0034367.1* and *CSS0012962.1* (Chr7 and Chr2), respectively. Ultimately, 136 full-length genes with high-confidence MADS domains were discovered in the tea plant genome, including 63 type II genes and 73 type I genes based on sequence similarity (Appendix A). 

### 2.2. Maximum Likelihood Phylogenetic Analysis of MADS-Box Genes

The maximum likelihood phylogeny of the type II MADS-box protein sequences of tea plant, grape, and *Arabidopsis* showed that 63 genes from the tea plant could be assigned into 14 major MIKC-type gene clades, with the members numbers described in square brackets: *AGAMOUS-LIKE6* (AGL6) [1], *SEPALLATA1* (*SEP1*) [3]/*SEP3* [2], *APETALA1* (*AP1*) [1]/*FUL* [6], *SUPPRESSOR OF OVEREXPRESSION OF CONSTANS1* (*SOC1*) [8], *B-sister*(BS) [1], *APETALA3* (*AP3*) [1]/*TOMATO MADS BOX GENE6*(TM6) [2]/*PISTILLATA* (*PI*) [2], *TM*8 [1], *SHORT VEGETATIVE PHASE* (*SVP*) [4], *AGL12* [1], *AGL15* [2], *ANR1* [11], *AGAMOUS* (*AG*) [3]/*SEEDSTICK* (*AGL11*) [1], *FLOWERING LOCUS C* (*FLC*) [3], and *MIKC^*^*(*Mδ*) [10] (Figure 1). The *ANR1*, *MIKC**, and *SOC1* subfamilies had the largest numbers of *CsMADS* genes, while *AGL11*, *TM8*, and *AGL12* had the smallest number, with only one copy each. Except for the *TM8* and *TM6* subclades known from other flowering plants, all type II genes in the tea plant had orthologous genes from *Arabidopsis.* In addition, the numbers of AP3/TM6/PI, AP1/FUL1, SOC1, ANR1, and MIKC* subfamilies in the tea plant were much higher compared to their *Arabidopsis* and grape counterparts. Type I genes in the tea plant could be divided into two subclasses: Mα (39) and Mγ (34). No gene member was found in Mβ, which was similar to the results in grape and orchid [43,44]. Phylogenetic analysis of type I genes from grape, *Arabidopsis*, and the tea plant showed that many genes appeared to be unique to the tea plant, suggesting these genes may have undergone independent replication and evolution (Appendix A).

### 2.3. Analysis of Genes Structure and Conserved Motifs

We analyzed the conserved motifs for all full-length MADS-box genes, as well as their exon-intron composition (Figure 2). Overall, the composition of motifs in the same subfamily was relatively conserved. Most of the MIKC^C^ genes shared six motifs (motifs 8, 1, 4, 6, 2, and 5), among which motif 1 represents the MADS-box domain. However, in one case of the AP3/TM6/PI subclade, *TM6-like* genes were characterized as having only three motifs (motifs 1, 2, and 8), indicating that their protein structures diverged from those of other B-class members during evolution. The distribution of specific conserved motifs varied among the different subfamilies. For example, all three members of the FLC subfamily contained no motif 6. Unlike MIKC^C^, MIKC*-type genes contained fewer motifs (only three to four). Ten MIKC*-type genes with identifiable K domains could not be detected in pfam or prosite. Motif 3 and motif 9 were exclusive to Mα and Mγ, respectively. The structure of type II genes was longer and more complex than that of other genes due to the presence of multiple exons, while type I genes had only one or two exons. In contrast to the previous reports that nearly half of MIKC^C^ genes lacked introns in rice, *Arabidopsis*, and bamboo [45], homologs in the tea plant tended to retain more introns (about five to eight). Exceptions to this rule included two newly identified genes (*CsFUL1d* and *CsFUL1e*) and a known gene, *CsFUL1c*, whose intron numbers were extremely low, with only one intron each. Notably, we observed that a few genes (e.g., *CsFLC3*, *CsANR1a-1*, and *CsANR1a-2*) had extremely long introns (20 kb~60 kb). To date, such patterns have only been reported in a few species, such as bamboo and wheat [46,47].

### 2.4. Chromosomal Location and Gene Duplication Events of the MADS-Box Gene Family in Tea Plant

To explore the genomic distribution of the MADS-box gene family, we visualized the chromosomal positioning of the *CsMADS* genes according to genome annotation information (Figure 3). Tea MADS-box genes were unevenly spread across all chromosomes. Chr5 did not contain the MADS-box gene, while Chr2 and Chr11 contained the highest and lowest numbers of the gene, with twenty-seven and one, respectively. Owing to established gene nomenclature, type II genes from the same subfamily were clearly prone to duplicate within different chromosomes to realize full functionality. However, type I genes were often found to be evolutionarily closely related to each other, exhibiting a cluster in the physical map. For example, fifteen Mγ subfamily members were adjacent to Chr2 and five Mα members on Chr3. Moreover, eleven genes were still anchored to unattributed scaffold fragments, and five newly identified MIKC-type genes were located on Chr15, Chr13, Chr7, Chr3, and Chr10. 

To clarify the probable relationship between *CsMADS* genes and potential duplication events within the genome, both the occurrence of tandem duplication and large-scale segmental duplication during evolution were analyzed using a genome synteny analysis (Figure 4 and Appendix A). A total of ten pairs of paralogous genes emerged from tandem duplications (TD), with nine of them from type I. Only one pair (*CsANR1a-1* and *CsANR1a-2*) was observed from type II, located within a 200 kb chromosomal region. Several ancient tandem arrangements were seemingly retained in the tea plant, including *CsSEP3b* and *CsFLC1* on Chr 1; *CsSEP3a*, *CsFLC3*, and CsFUL1c on Chr 10; *CsSEP1c* and *CsFUL2* on Chr 2; and *CsAGL6* and *CsSOC1a-3* on Chr 3. Such patterns have been previously reported in grapes, potatoes, peaches, and other species [48,49,50]. The tea genome contained 18 pairs with 34 *CsMADS* genes originating from WGD or segmental duplication. The largest number of MADS-box genes with collinearity was found on Chr 2 with five, followed by Chr 1 and Chr 9. Unlike tandem duplications, 88% (30/34) of segmental duplicated genes belonged to type II, while only two collinear gene pairs were from type I, namely, *CsMADS1A3i*/*CsMADS1A3h* and *CsMADS1G1b*-8/*CsMADSG1b*-9. Ultimately, segmental duplication played an important role in the expansion of type II genes, whereas tandem repeat duplication mainly contributed to type I gene expansion.

To better understand the nature and extent of selection constraints acting on the tea plant MADS-box gene family, the non-synonymous (Ka) and synonymous (Ks) divergence values of 18 MADS-box gene pairs were calculated. For 12 segmental duplicated *CsMADS* gene pairs with valid Ks values, all Ka/Ks values of the corresponding MADS-box gene pairs were less than 1 (Appendix A), suggesting that the tea plant MADS-box gene family might have experienced negative or purifying selective pressure during evolution [51,52]. This result shows that five gene pairs diverged about 27–43 million years ago (MYA), and four gene duplication events occurred around 100 MYA (Appendix A).

### 2.5. MADS-Box Gene Expression Analysis among Different Tissues and Seasons

To explore the expression profiles of tea plant MADS-box genes, we analyzed the RNA-seq data of 136 gene members in different tissues and seasons. Log2-transformed expression values of type II and type I genes were visualized using a hierarchically clustered heatmap, (Figure 5A,B, Appendix A). According to expression similarities, sixty-three type II genes were categorized into three major clusters (A-C) and eight expression modules (I-VIII) (Figure 5A). Cluster A was further assigned to expression modules I and II. The genes in this cluster showed flower-specific expression with the majority of putative floral homeotic genes clustered together. Gene expressions in module I (*CsAP1*, *CsAP3*, and *CsTM6a*) were also detected in vegetative tissues. All members from the SEP1/SEP3, PI, AGL11, AG, BS, and AGL6 clades were clustered in module II and showed exclusive expression in flowers. The overall expression levels of genes in cluster B were relatively low and could be further divided into III, IV, and V expression modules. Among them, module III comprised three *ANR1-like* (*CsANR1a-1*, *CsANR1a-2*, and *CsANR1c*) genes and one *FUL-like* (*CsFUL2*) gene. These *ANR1-like* genes were specifically expressed in roots, while *CsFUL2* was expressed during both root and flower development. In modules IV and V, most genes were undetectable or expressed at very low levels, including most members of the *ANR1* and *MIKC** subfamilies. However, two MIKC* genes (*CsMADSD1a* and *CsMADSD1b)* were specifically expressed in autumn flowers, and *CsTM8* in module IV was specifically expressed in the stems. Module VI included four genes from three subfamilies, *CsFUL1a*, *CsFLC2*, *CsFLC1*, and *CsSVP3a*, which exhibited extensive expression patterns throughout the annual cycle, with high transcript accumulation. Most of the *SOC1-like* genes were assigned to module VII, and together with two SVP members (*CsSVP2* and *CsSVP3b*), were widely expressed in non-floral tissues. The VIII expression module could be detected in all organs, but the overall expression level was low. 

Generally, gene expression patterns from the same subfamily were relatively conserved, although expression levels of specific family members could vary across tissues. Notably, the expression patterns of five members of *AP1/FUL-like* genes (*CsAP1*, *CsFUL1a*, *CsFUL1b*, *CsFUL1c*, and *CsFUL2*) were very diverse and present in four expression modules. Transcript level changes of some genes specific to seasons were also observed in this study. For example, the expression of *CsAGL6* reached the highest level in spring flowers, followed by winter and autumn flowers. The same pattern was also detected in *CsAP1*, *CsFUL2*, *CsAGL11*, *CsBS*, and *CsSEP3a*, while *CsSEP1a-c* and *CsSEP3b* presented an opposite expression pattern. Notably, the transcript abundances of *CsFUL1a* and *CsFLC2* were significantly more upregulated in winter than in other seasons. In addition, *CsSVP3b* was preferentially expressed in the vegetative organs of autumn, while *CsSVP3a* tended to be expressed in winter and spring samples. The expression preferences of members from the same subfamily in different seasons and different tissues suggested expression differentiation and functional divergence after gene duplication. Most type I genes had very low or no detectable expression (Figure 5B). This result is consistent with previous studies showing that type I gene expression is restricted to specific cells and tissues, such as female gametophytes and developing seeds [53].

### 2.6. MADS-Box Gene Expression Analysis in Response to Cold Stress

To further study the role of MADS-box genes in response to cold stress, the expression profiles of MADS-box genes were obtained in tea leaves using transcriptome assembly (Appendix A). We assembled two datasets of low-temperature-related transcriptomes, one for short-term cold stress (Figure 6A) and the other for long-term cold acclimation and de-acclimation (Figure 6B). In the results, some MIKC-type genes showed cold-responsive expression patterns. The expression of *CsFUL1a* in one bud and two leaves was significantly up-regulated by low temperatures. Consistent with this result, *CsFUL1a* expression was remarkably increased during the 7 d chilling acclimation stage (CA2_7d) and then inhibited after recovering from cold acclimation. However, the expression trend of *CsFUL1b* was diametrically opposite. *CsTM6a* did not exhibit differential expression under cold treatment, but its expression level significantly increased after de-acclimation. *CsSVP1*, *CsMADSD1f*, and four *SOC1-like* members (*CsSOC1a-1*, *CsSOC1a-2*, *CsSOC1a-4*, and *CsSOC1a-5*) were also up-regulated during the de-acclimation stage. *CsFLC1* was slowly downregulated with continued cold acclimation, even after turning warm, while *CsFLC2* was up-regulated by cold stress. Except for the aforementioned genes that respond to low temperatures, the remaining genes suffered from an expression that was either too low or not sufficiently different from that of the control under cold stress.

### 2.7. A Putative Floral ABCE Model for Tea Plant According to Expression Profiles

The expression patterns of twenty-one ABC(D)E model genes were detected in four floral organs and three developmental stages of flower based on transcriptome data (Figure 7A, Appendix A). Among the five detected *AP1/FUL-like* gene members, only *CsAP1* and *CsFUL1b* were involved in both sepal and petal specification, and *CsFUL2* was expressed in sepals but at a low level. All five B-class genes (*CsAP3*, *CsPI1*, *CsPI2*, *CsTM6a*, and *CsTM6b*) from the tea plant displayed broad expression patterns. In addition to the expected second and third whorls, the expression domains of these B-class genes extended into the first and fourth whorls. For C-class genes, the transcripts of *CsAG2a* and *CsAG2b* strongly accumulated in pistils and stamens. Another *AG-like* gene, *CsAG1*, was preferentially expressed in stamens, indicating the stamen identity of the C-function. The E-class genes *CsSEP3a*, *CsSEP3b, CsSEP1b,* and *CsSEP1c* were expressed throughout four whorls and may represent candidate E-function genes. *CsSEP1a* may instead exert a partial E-function since it showed a relatively high expression in sepals but no expression in stamens. Notably, *CsAGL6* was mainly expressed in sepals and petals and may constitute the A-function together with *CsAP1* and *CsFUL1b*. A putative ABCE model was proposed according to the expression profiles and deduced functions of A-, B-, C-, and E-class genes in four whorls (Figure 7B). In total, 16 ABCE model-related genes were selected for RT-qPCR analysis to validate the reliability of the RNA-seq results (Figure 8). The RT-qPCR results strongly agreed with our RNA-seq results.

## 3. Discussion

MADS-box genes comprise a large family of transcription factors that are widely present in all flowering plants. Due to the availability of the recently published tea plant genome sequence, *CsMADS* genes can now be systematically identified and analyzed [42]. In this study, we searched for MADS-box genes in the tea plant genome using a self-built nucleic acid HMM, resulting in the identification of 136 family members. The total number of identified MADS-box genes was approximately two times that of a previous study [42]. Highly contiguous genome sequencing of the tea plant using a practical and delicate strategy led to a significant increase in gene numbers. A nucleic acid HMM search was sufficient to detect all potential MADS domains in genomic regions, including 124 known and 27 novel genes. Manual curation of the gene annotations greatly improved the accuracy of the gene model set for MADS-box sequences using assembled transcriptome datasets and homology predictions. 

### 3.1. Evolution of the MADS-Box Family in Tea Plant

The tea plant presented a relatively high number of type II genes (63), with the number in many flowering plants ranging from 40 to 70 [54]. According to the phylogenetic analysis, tea plant type II MADS-box genes were distributed in all 14 major eudicot subfamilies, which is consistent with the classification and results for *Arabidopsis* and grape [18,55]. The presence of all relevant genes in these subfamilies reemphasized the completeness of our identification results. Different retention histories for duplicates in the different clades were observed in the tea plant, *Arabidopsis*, and grape (Figure 1). *TM6-like* and *TM8-like* genes are not found in *Arabidopsis*, but are present in tea plants and grapes. Previous findings indicated that independent gene loss occurred twice in the lineage leading to *Arabidopsis* [17,56]. Moreover, members of the AP3/TM6/PI, AP1/FUL, SOC1, ANR1, and MIKC* subfamilies were overrepresented in the tea plant, while *Arabidopsis* and grape contained more genes in the FLC and SVP clades. The presence of increased homologs from these clades might help fulfill specific functions, such as flower organization, flowering time, and root and gametophyte development. 

Gene duplication events played a critical role in the expansion of the tea plant MADS-box gene, as observed in other species belonging to different taxonomic groups [57]. Based on genome synteny analysis, 30 paralogs composed of 53 genes were identified, of which 30 belonged to segmental duplications, and 23 were tandem duplication events. Our results showed that the expansion of type II MADS-box genes in the tea plant might have been primarily caused by WGD/segmental duplication, whereas type I genes were mostly derived from tandem duplications in individual chromosomes. Indeed, it has been suggested that several whole genome duplication events gave rise to most type II MADS-box genes, while type I genes are generally associated with smaller-scale duplication events [57]. The tea plant genome acquired an additional WGD after WGT-γ, which might have been responsible for the increased number of type II genes.

### 3.2. Extensively Conserved Function of MADS-Box Genes in Tea Plant Development

The gene expression patterns of most MADS-box genes in the tea plant showed extensive similarities with orthologous genes described in model plants such as *Arabidopsis* and related grape species, indicating that the functions of MADS-box genes were widely conserved among these species. For example, most tea plant MADS-box genes specifically expressed in flowers (*SEP1/SEP3-like*, *PI-like*, *AGL11-like*, *AG-like*, *BS-like*, and *AGL6-like* genes) were putative floral homeotic genes. Like their *Arabidopsis* and grape counterparts, *ANR1-like* genes in the tea plant presented the highest expression in roots. *CsAGL12* was mainly expressed in roots and flowers, which is consistent with the roles of the *AGL12* homologous gene (*AtXAL1*) in *Arabidopsis* in regulating root development and flowering transitions [58]. Tea plant *SOC1-like* genes showed expression patterns only in vegetative organs, which was consistent with the observations in *Arabidopsis* and grape, where the major expression domains of *SOC1-like* genes are not the floral organs [59]. Some *FLC-like* and *SVP-like* genes from the tea plant were ubiquitously expressed throughout the plant’s life cycle in different tissues and might be involved in several developmental processes, as observed in *Arabidopsis*. The gene expression patterns were generally conserved among most subfamilies. However, diverse expression patterns of *AP1/FUL*-like genes were observed in the tea plant, suggesting that these homologous genes might be differentiated into multiple functions. 

### 3.3. The Involvement of Tea Plant MADS-Box Genes in Cold Stress Responses

Previous research has indicated that some MADS-box genes play an important role in cold stress [60,61,62,63]. In this study, one *FUL1-like* gene *(CsFUL1a)* was strongly induced during the cold acclimation stage and showed significantly high expression in all winter tissues, which might be involved in the adaptation of tea plants to the severity of winter and flowering regulation. Apart from the *FUL1-like* gene (*TM4*), low temperatures can simultaneously increase the transcript levels of *TM5* (*SEP*), *TM6* (*DEF*), and *TAG1* (*AG*) in tomatoes. These genes may be involved in abnormal flower production at low temperatures [37,64,65]. The expression of these orthologs in tea plants exhibited no significant changes under cold stress, and *CsTM6a* was up-regulated after de-acclimation, indicating differential stress responses from tomatoes. Another gene strongly induced by cold acclimation was *CsFLC2,* the highest expression of which was detected in winter bud. Coincidently, four *SOC1-like* members (*CsSOC1a-1*, *CsSOC1a-2*, *CsSOC1a-4*, and *CsSOC1a-5*) preferentially expressed in summer were repressed under cold stress but up-regulated during the de-acclimation stage. The antagonistic expression patterns of *FLC* and *SOC1* fit well with the fact that *FLC* was a negative regulator of *SOC1* during flowering transitions. *SOC1* was involved in cross talk between cold response signaling and flowering regulation by directly repressing *C-Repeat Binding Factor* (*CBF*) genes, which could activate both *Cold Regulated* (*COR*) genes and FLC expression [66,67]. Moreover, in kiwifruit, SVP homologs from the SVP2 and SVP3 clades may act as suppressors during bud dormancy [68,69]. The expression of dormancy-associated *PpDAM5* and *PpDAM6* in peaches (*Prunus persica*) from the SVP clade was up-regulated in autumn and down-regulated by cold in winter [69,70]. Its ortholog, *CsSVP3b,* exhibited the highest expression in all autumn samples, and *CsSVP3a* was repressed by short-term cold stress, highlighting the potential involvement of these two genes in bud dormancy. Taken together, both seasonal expression analysis and cold treatment expression analysis suggested that these cold-responsive genes might play diverse roles in seasonal low-temperature-dependent developmental processes. 

### 3.4. A Conserved and Ancient Floral ‘ABCE’ Model in Tea Plant

Four whorls of organs (sepal, petal, stamen, and pistil) from three different flower developmental stages were collected to investigate the regulatory relationships between ABCE homologs and floral organ development using RNA-seq and qRT-PCR (Figure 8). Our results demonstrate that the expression patterns of these genes largely agree with those of the classical floral ‘ABCE’ model. Previous studies have shown that *AGL6*-like genes formed a ‘superclade’ together with *SEP1*- and *SQUA*-like genes and shared similar functions with genes from these two sister clades in specifying floral meristem and organ identity [71]. In this study, a AGL6 homolog gene, *CsAGL6*, tended to act as an A-function candidate gene because its expression pattern was similar to the patterns of A-lineage genes. Five B-class genes were highly expressed in petals and stamens, indicating conserved functions in controlling the development of stamens and petals. C-class genes (*CsAG1*, *CsAG2a*, and *CsAG2b*) displayed high expression levels in the stamens and pistils, which is in line with its expected functions in specifying both male and female reproductive organs [20]. Five E-class genes fulfilled full or partial E-function activities that redundantly determined floral organ identity. Moreover, compared with eudicot model systems, a wider range of expression domains was observed in the tea plant. Notably, the expression domains of five B-class genes were expanded into sepals and pistils. This phenomenon is used to refer to the ancient ‘ABCE’ model mainly found in lineages that diverged early, such as *Aquilegia coerulea* [72], waterlilies [73], and magnoliids [74]. Interestingly, an expanded expression domain of B-function genes was also observed in other Camellia plants, *Camellia japonica* and *Camellia changii Ye*, which was thought to be responsible for petaloid phenotype and double flowers in these two species [75,76]. Thus, we speculate that this phenomenon may have occurred as early as in the ancestor of Camellia plants.

## 4. Materials and Methods

### 4.1. Identification of MADS-Box Genes and Curation of a Gene Model

The tea plant genome was retrieved from the TPIA website (http://tpia.teaplant.org/index.html) [77]. The Hidden Markov Model (HMM) of the SRF (MADS) family and (K-box) (PF00931 and PF01489) were downloaded from the Pfam website (http://pfam.sanger.ac.uk/ (accessed on 8 January 2021)) [78]. In addition, hmmsearch in the HMMER v3 software [79] was used to scan the tea plant protein sets for genes containing these two domains. A high-quality protein dataset was obtained through preliminary filtering based on the e-value and length. MAFFT v7.215 [80] was used for multiple sequence alignment of the conserved regions. The nucleic acid sequences corresponding to the aligned MADS-box and K-box domains were extracted and converted into a tea-specific nucleic acid HMM using the hmmbuild function of the HMMER software. This constructed HMM was used to scan the tea plant genome sequence by executing the “nhmmer” command to obtain sites containing this domain (Appendix A). 

For the genome-wide candidate sites obtained in the last step, regions without corresponding mapped genes were considered potential new functional sites. Subsequently, Genewise [81] was used to screen the genomic regions near the new sites by inputting homologous genes from grape, kiwifruit, and other related species as a query. Gene sets of the floral organ transcriptome assembled in our laboratory (unpublished) were especially useful for the functional annotation of genes expressed in the flowers. The gene models of known genes without the MADS-domain or K-box domain were manually corrected using the same strategy. All full-length genes were retained for further analysis.

### 4.2. Sequence Alignment and Phylogenetic Analysis

To study the evolutionary relationships between the newly discovered MADS-domain genes in the tea plant, multiple sequence alignments were performed based on well-identified protein sequences of grape and *Arabidopsis* using MAFFT software [80] with 1000 bootstrap replicates. The MADS-box protein sequences in *Arabidopsis* and grape were obtained from the Plant TFDB website (http://planttfdb.gao-lab.org/index.php (accessed on 8 January 2020)) [82] and the research of *Grimplet* et al. [55], respectively. Maximum-likelihood phylogenetic trees were constructed using the maximum likelihood approach in FastTree v.2.1.10 software [83] with the GTR+CAT model. The phylogenetic tree was further refined using Evolview online tools (https://www.evolgenius.info/evolview/) [40].

### 4.3. Gene Nomenclature of Tea Plant MADS-Box Genes

To provide insight into the potential functional roles of relevant genes and facilitate future research, we renamed the MADS-box gene family members. For MIKC^C^ genes, we took the association and phylogenetic relationships of subfamilies into consideration and also referred to the gene nomenclature of orthologs in *Arabidopsis* and grape. Each gene begins with “*Cs*”, the abbreviation of the species name *Camellia sinensis*, followed by the name of the most important *Arabidopsis* gene in this subfamily. In some clades where no *Arabidopsis* orthologs exist, we took the nomenclature of the grape as a reference, such as the *TM8* clade, with the exception of *TM6*-like genes, which are only found in eudicots and named after *AP3*-like genes in previous grape research. In cases where several subclades appear in a given subfamily, and previous studies have failed to describe them, if the subfamily name ends with a number, then we added lowercase letters (a, b, c, etc.) to distinguish each subfamily. If the subfamily could be further divided, we added a dash (-) and a number after the letter; if the subfamily name ends with a letter, we added a number to distinguish. For genes assigned to MIKC* and type I groups, we named them after the grape genes. The naming of Type I genes was largely performed the same as that of MIKC* genes, except that the number “1” was added after “MADS”.

### 4.4. Characterization of the Gene Structure and Conserved Motifs of CsMADS

The available information for exons and introns was retrieved from the tea plant genome annotation file and then visualized using TBtools [84] with the full-length coding sequence (CDS) and genomic sequence. The online MEME suit software v 4.12.0 (http://meme-suite.org/tools/meme (accessed on 1 July 2015)) [85] was used to search conserved motifs in each observed CsMADS protein with the parameters as follows: maximum number of motifs, 10. The default values were employed for the other parameters. The motifs of MADS-box genes were annotated using the SMART program (http://smart.embl-heidelberg.de/ (accessed on 1 January 2009)).

### 4.5. Gene Mapping, Gene Duplication, Synteny, and Ka/Ks Substitutions of MADS-Box Genes in Tea Plant

According to the annotation file (GFF3) for the tea plant genome downloaded from the TPIA database (http://tpia.teaplant.org/index.html (accessed on 24 March 2020)), the positions of tea plant MADS-box genes on the chromosomes were determined, and Tbtools software [84] was used to map all MADS-box genes onto chromosomes. The Multiple Collinearity Scan toolkit X (MCScanX) [59] was applied to identify segmental and tandem duplications of *CsMADS* genes with default parameters, and the results of synteny were visualized using Circos software [60]. Synonymous (Ks) and nonsynonymous (Ka) substitution rates and their Ka/Ks ratios were estimated via the Simple Ka/Ks Calculator module in TBtools [84]. The divergence time was calculated according to the formula T = Ks/2r, where λ refers to the clock-like rate (6.5 × 10^−9^ synonymous substitutions per site per year) [86].

### 4.6. Gene Expression Analysis of Tea Plant MADS-Box Genes

To explore the expression patterns of tea plant MADS-box genes, raw RNA-seq datasets were mined from the NCBI database under the projects PRJEB39502, PRJNA411886, and PRJNA387105 (Appendix A). The project PRJEB39502 comprised nineteen tissue samples, including five tissues (root, stem, bud, leaf, and flower) in four seasons (flowers during summer excluded). The project PRJNA411886 comprised six treatments: FL0 h (control, 0 h), FL24 h (cold treatment, 24 h), and FL48h (cold treatment, 48 h) at the fish leaf phase; TAB0h (control, 0 h), TAB24h (cold treatment, 24 h), and TAB48 h (cold treatment, 48 h) at the two leaves and one bud phase. The temperature of the cold spell was set at 4 °C during the day and 2 °C at night. The project PRJNA387105 consisted of five treatments: CK (25~20 °C, control); CA1_6h (cold acclimated under 10 °C, 6 h); CA2_7d (cold acclimated under 10~4 °C, 7 days); CA2_7d (cold acclimated under 4~0 °C, 7 days); DA_7d (25~20 °C, 7 days). Detailed descriptions of these transcriptome datasets were published by Wang et al. [6], Hao et al. [87], and Li et al. [88], respectively. In addition, flower expression in the four whorls of the three developmental stages (the green bud stage, deglazing stage, and heyday stage) was based on unpublished transcriptome data from our laboratory (Appendix A). The transcripts per million (TPM) values of tea plant MADS-box genes were calculated using a self-built script according to the HISAT2 and Stringtie pipeline [89]. The heatmaps were visualized via Morpheus (https://software.broadinstitute.org/morpheus) based on the transformed data of the log2 (TPM+1) values. 

### 4.7. Plant Materials and Quantitative PCR Analysis

Tea flowers at the white bud stage were collected from tea bushes (*Camellia sinensis* var. Rougui) cultivated at the tea garden of Fujian Agriculture and Forestry University in the Fujian Province of China (26°04′ N, 119°14′ E) under natural conditions. Floral organs (sepals, petals, stamens, and pistils) were separated by hand and immediately inserted into liquid nitrogen for freezing and preservation. Floral organs from at least five randomly selected individuals were harvested as one replicate, and three biological replicates were used for each sample.

In this study, a polysaccharide and polyphenol kit (Tiangen Biotech, Beijing, China) was used to extract the total RNA of floral organs, according to the manufacturer’s instructions. Total RNA was reverse-transcribed using EasyScript one-step gDNA Removal and cDNA Synthesis SuperMix (Transgen Biotech, Beijing, China). Genomic DNA was digested by DNase prior to reverse transcription, and the obtained cDNA template was diluted to a uniform concentration of about 100 ng/μL. Real-time PCR amplification was performed on a CFX96 Touch Deep Well system (Bio-rad, Hercules, CA, USA) using TransStart Top Green SuperMix (Tiangen Biotech, Beijing, China) with SBYR green fluorescent dye. The PCR mixture in a 20 µL volume contained 10 µL SuperMix, 0.4 µL forward primers, 0.4 µL reverse primers, 1 µL cDNA, and 8.2 μL ddH_2_O. The thermal cycling conditions were set as follows: 95 °C for 30 s; 40 cycles at 95 °C for 5 s and 60 °C for 34 s; and 95 °C for 15 s, 60 °C for 1 min, and 95 °C for 15 s. A housekeeping gene, *GADPH*, was used as an internal control for data normalization. All reactions were run in technical triplicate and biological triplicate. The 2^−ΔΔCT^ method (also called the comparative CT method) was used to calculate the relative gene expression levels. Details for the primers are provided in Appendix A.

## 5. Conclusions

In this study, a tea-specific nucleotide acid HMM was developed to identify 136 MADS-box genes in the tea plant genome, including 63 genes of type II and 73 of type I. In total, 27 novel genes were identified in this study for the first time. In-depth artificial curation largely improved the accuracy of gene models based on assembled transcriptome datasets and homology predictions. Seasonal expression analysis and cold treatment expression analysis suggested these cold-responsive genes might play diverse roles in seasonal low-temperature-dependent developmental processes. Moreover, the expanded expression domain of ‘ABCE’ model genes indicated that the tea plant has a conserved and ancient ABCE floral model. Overall, this work contributes to our understanding of the evolutionary history of the MADS-box gene in the tea plant and provides perspectives on corresponding functional analyses. 

## Figures and Tables

**Figure 1 plants-12-02929-f001:**
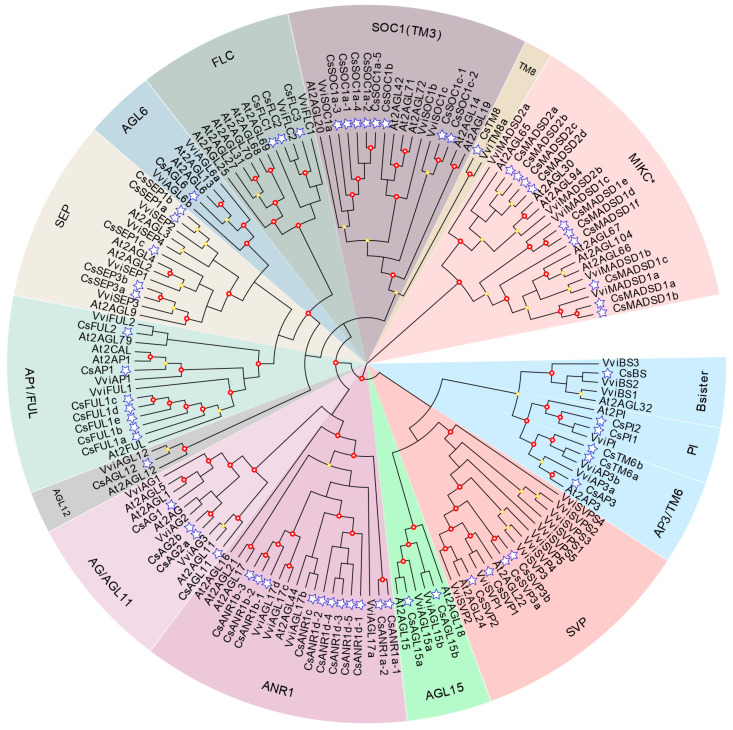
Phylogenetic relationship of type II MADS-box proteins. The maximum likelihood (ML) phylogenetic tree of MADS-box proteins from tea plant, *Arabidopsis*, and grape was constructed using FastTree with the GTR+CAT model. Tea plant genes are marked with blue stars. The subfamilies are presented in different colors. At, *A. thaliana*; Vvi, *Vitis vinifera*, Cs, *C. sinensis*.

**Figure 2 plants-12-02929-f002:**
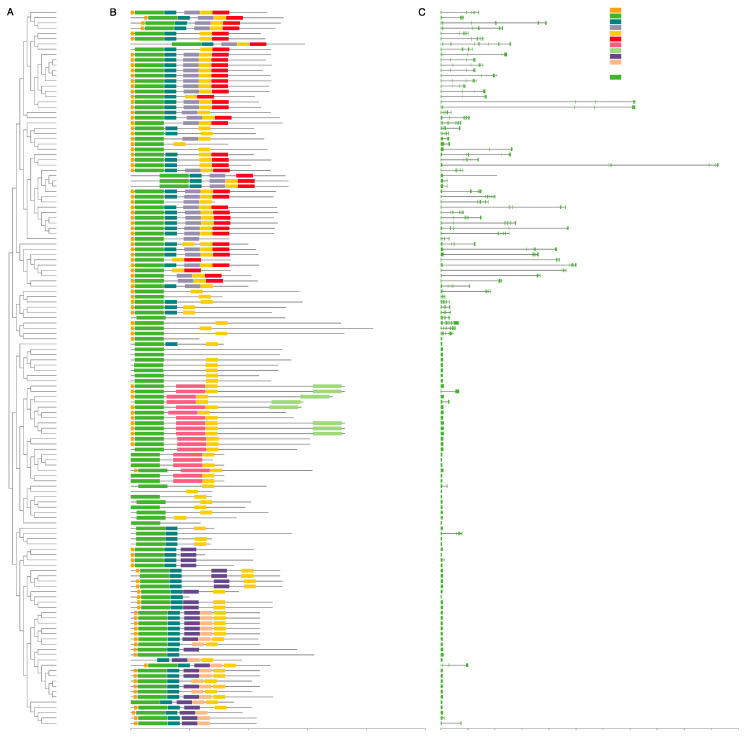
Phylogenetic relationship, gene structure analysis, and motif compositions of MADS-box genes in the tea plant. (**A**) A total of 136 full-length sequences of CsMADS protein were used to build a neighbor-joining tree, with 1000 bootstrap replicates. (**B**) Conserved motif patterns were identified in tea plant MADS-box proteins using the MEME webserver. Different motifs are represented by different colors. (**C**) The exon–intron structures of *CsMADS*. Green blocks represent exons, and gray lines indicate introns. The size scale is indicated at the bottom.

**Figure 3 plants-12-02929-f003:**
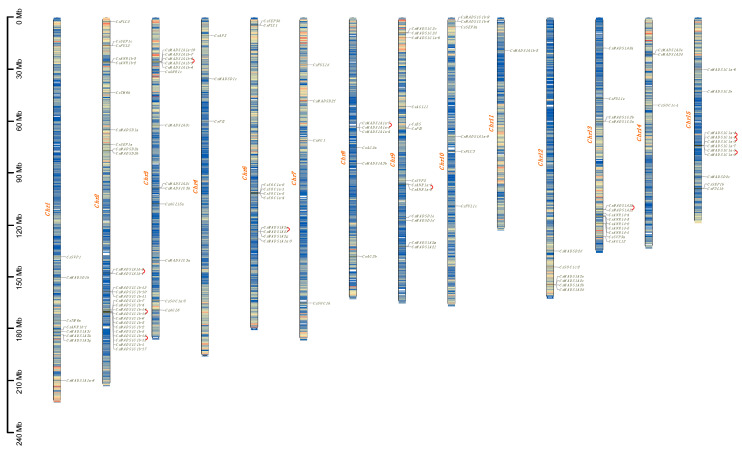
Chromosomal locations for the tea plant *MADS-box* gene family. Genes were mapped onto 14 chromosomes. The gene density of each chromosome was calculated with TBtools and is visualized by gradient colors from blue (low level) to red (high level). Tandem duplicates are linked with red curves.

**Figure 4 plants-12-02929-f004:**
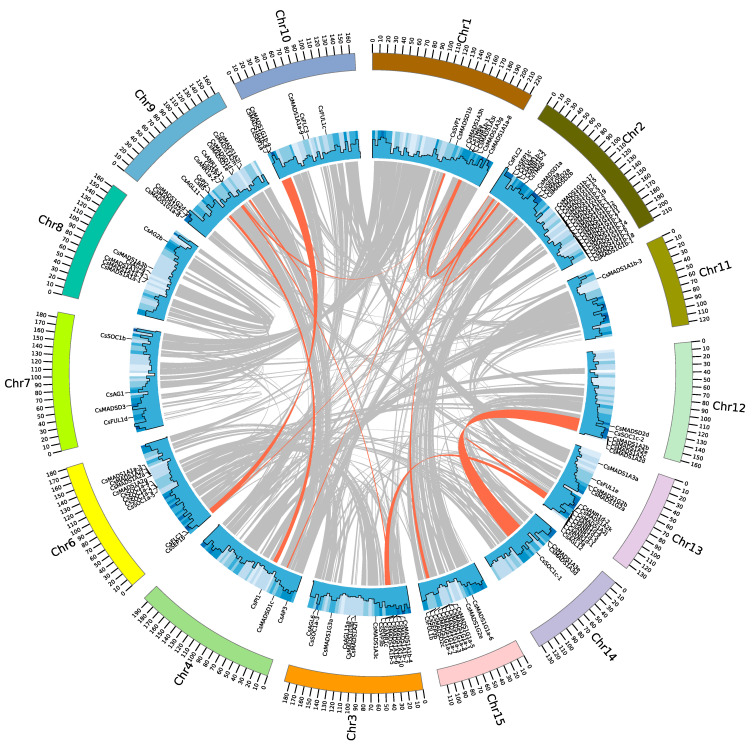
Synteny blocks of *CsMADS* genes in tea plant chromosomes. Gray lines indicate syntenic gene arrangements in the genome of the tea plant, while orange lines link gene pairs derived from segmental duplications or whole genome duplications.

**Figure 5 plants-12-02929-f005:**
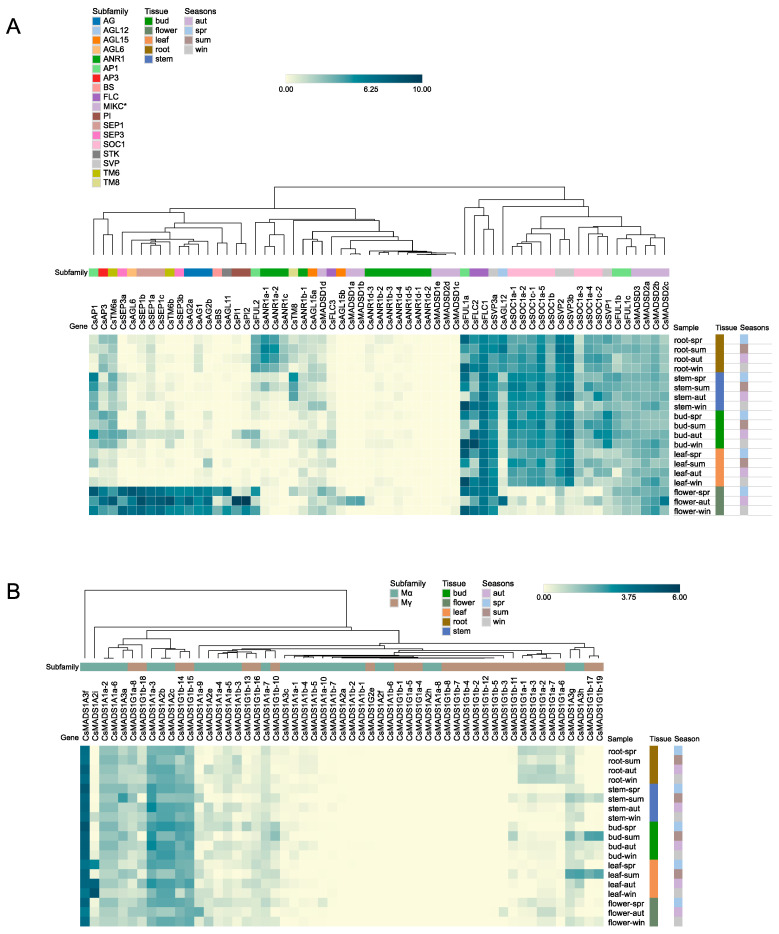
Expression patterns of tea plant MADS-box genes in different tissues (root, stem, bud, leaf, and flower) and seasons (spring, summer, autumn, and winter). (**A**) Clustering heatmap of type II *CsMADS* expression profiles. The three major expression groups are marked as A, B, and C, and roman numbers below/above the horizontal line represent further subdivided modules. (**B**) Clustering heatmap of type I C*sMADS* expression profiles. Genes and tissues are listed in Appendix A. The TPM values were adopted for the relative gene expression levels and were normalized by genes (log2TPM). The color scale bar is provided at the top-left corner.

**Figure 6 plants-12-02929-f006:**
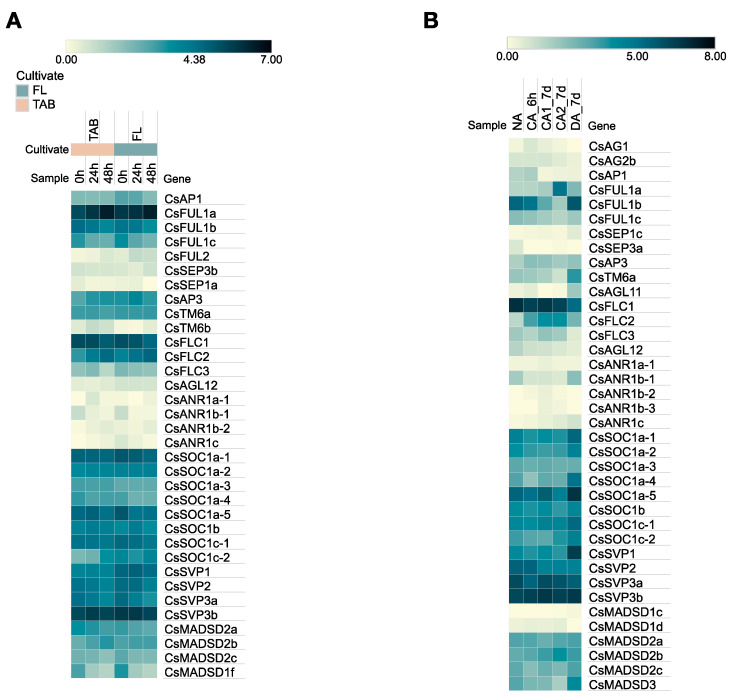
Expression analysis of *CsMADS* in response to cold stress. (**A**) Expression analysis of *CsMADS* in response to cold stress in fish leaf (FL) and two-leaf-and-one-bud (TAB) tissues. (**B**) Expression analysis of *CsMADS* in response to cold acclimation. The TPM values were adopted for the relative gene expression level and normalized by gene (log2TPM). The color scale bar is located at the top.

**Figure 7 plants-12-02929-f007:**
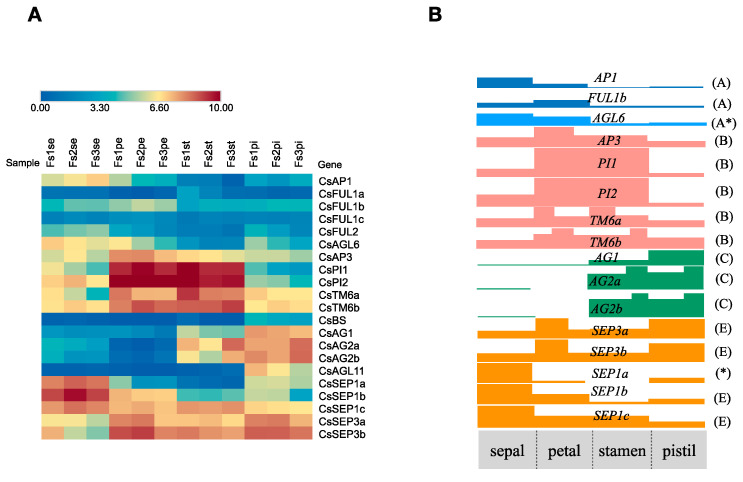
Expression profiles of ABC(D)E genes in the four whorls of the flower and a proposed floral ABCE model for the tea plant. (**A**) Heatmap of expression patterns of tea plant A-, B-, C-, D-, and E-class genes in floral organs (sepals, petals, pistils, and stamens) of three developmental stages. “FS” stands for flower stage, “se” for sepals, “pe” for petals, “pi” for pistils, and “st” for stamens (**B**) A floral model for the tea plant built according to the expression patterns described in Figure 7A and the expected functions of detected MADS-box genes. “*” represent that putative gene function was different from expectation due to expression split.

**Figure 8 plants-12-02929-f008:**
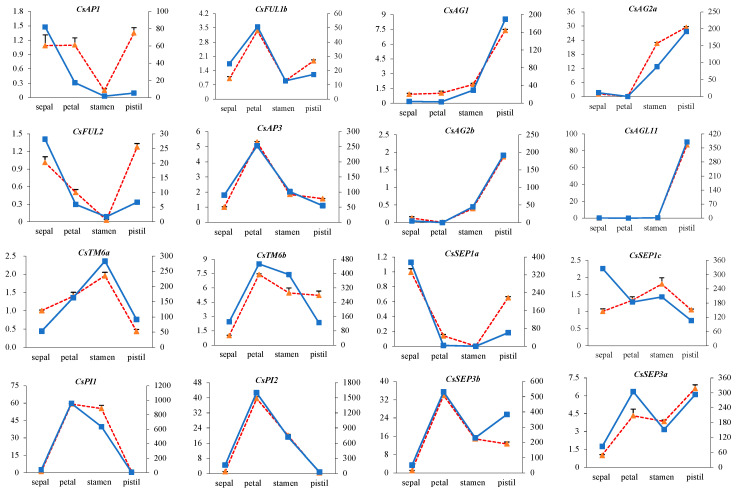
RT-qPCR verification of 16 ABCE model genes. The left y-axis represents the relative expression of the RT-qPCR results and the right y-axis stands for the FPKM value from the RNA-seq results. The blue solid line represents the TPM value, and the orange dashed line is the relative expression.

## Data Availability

All data in this study can be found in the manuscript or the Appendix A.

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
