# Peer review of "Structural and Functional Analysis of the MADS-Box Genes Reveals Their Functions in Cold Stress Responses and Flower Development in Tea Plant (Camellia sinensis)"

_plants, 2023, doi:10.3390/plants12162929_

Round 1

Reviewer 1 Report

1. English is poorly written and lot of grammatical errors are found in the manuscript. Authors should improve the manuscript.

2. Authors should maintain a uniformity in writing. Write down either the botanical or

   common name of plants in the entire manuscript.

3. The quality of all the figures is very poor. It should be improved.

4. Repetition should be avoided throughout the manuscript.

5Interaction analyses with other proteins would further add the value in the study. You may go through the ‘Investigation of Roles of TaTALE Genes during Development and Stress Response in Bread Wheat’, and follow for interaction study.

6. Discussion section is poorly written and bulky. Only Key points should be discussed.

7. What is the reason behind the occurrence of variable number of introns in the genes of same

      Family?

8. Introduction lacks several updated info including, ABCDE model of flowering in Phalaenopsis

9. Since it is mostly in-silico work author should include evolutionary analysis. They may follow the the article ‘Molecular Characterization, Evolutionary Analysis, and Expression Profiling of BOR Genes in Important Cereals’

I could not see any statistical analyses in the RT PCR data. Further, usually two internal control are being used now a day, authors have used only one here, and why GAPDH is used, not actin?

Extensive editing required

Author Response

All the comments have been addressed in the Rebuttal letter. 

Reviewer 2 Report

Comments and Suggestions for Authors

Dear Author,

I have an honor to review the manuscript entitled “Structural and Functional Analysis of the MADS-Box Genes in Tea Plant Reveals Their Functions in Cold Stress Response and Controlling Flower Development” a research article submitted to MDPI Journal, Plants. Authors of this manuscript analyzed evolutionary relationship and functions of MADS-Box Genes in tea plant using various bioinformatic tools. Overall, the experiments are performed well and the results are convincing. Thus, the presented results takes up an important topic consistent with the profile of the Journal.

-However, I have some suggestions, which might improve the manuscript to make important to the wider reader.

-Few suggestions I have mentioned in the main text pdf file. Please check

-Some major and other minor comments are as below

-There are many places where grammar can be improved. I suggest a careful revision by an expert. I've just noted a few here. No line number, so difficult to comment

Title: title could be improved. Suggestion, “Structural and Functional Analysis of the MADS-Box Genes Reveals Their Response in Cold Stress and Flower Development in Tea Plant”

Abstract: -Good organization with results order.

- Here, 136 MADS-box genes were detected from reference genome of tea plant (Camellia sinensis) by employing a 569 bp HMM developed using nucleotide sequence, including 73 type I and 63 type II genes------------Difficult to understand. Make meaningful with good grammatical structure. Need elaboration of HMM, MIKC at first time use

- Expression profiles of tea plant MADS-box genes in different tissues and seasons were analyzed-------what is the significant relationship between tissue and seasons?

-Keywords: -Better use alphabetic order

1. Introduction:

-Introduction is informative enough

- Due to the highly repetitive, heterozygous and relatively large (approximately 3G) genome of tea plant, as a perennial woody plant, it is challenging to perform genome assembly--- “as a perennial woody plant” does not match with this statement. Also need reference

-In introduction “type II (called MEF2-like or MIKC-type)”

-In abstract “including 73 type I and 63 type II genes”

If type II is MIKC-type, then what does this mean “An additional 27 genes were identified with 5 genes from MIKC-type”.

- Mβ, or Mδ

2. Results

-Database browsing time is some what important for wider reader

-Fig. 2. A, B, C are not present in the fig.

-To explore the expression profiles of tea pant MADS-box genes, we analyzed RNA[1]seq data of 136 gene members in different tissues (roots, stems, leaves, flowers, buds) and in different seasons (spring, summer, autumn, winter)---program name?

-Figure or Fig. make uniform throughout the text

-A total of 21 ABC(D)E model genes were selected to further investigate their roles in floral organ development in tea plant.- ---What was the basis for selection of only 21 from 136

-From 21 genes 16 ABCE model-related genes were selected for RT-qPCR, why only 16 but not all 21?

-Also need discussion about RT -PCR experiment in tea and many other plants

Fig. 8. Resolution is very low. Need more indication in the fig description. Mention Fs2se……all. SE result from how many experiments? Experimental/Biological replication?

4. Materials and Methods

4.6. Gene Expression Analysis of Tea Plant MADS-box Genes

4.7. Gene Expression Analysis of Tea Plant MADS-box Genes

Both are same title

- What are the floral organs have been used and what was the sampling process?

 Minor editing of English language required

Author Response

The comments have been in addressed in the rebuttal letter.

Round 2

Reviewer 1 Report

Ms is improved and can be now accepted.

Author Response

The authors are thankful for reviewer's valuable comments and accepting our response. 

Reviewer 2 Report

Manuscript has been improved largely by the authors following comments and suggestions

However, Database browsing time is somewhat important for wider reader. You may incorporate in the text

The supplied pdf file is not readable completely due to application of "review function" and converted to pdf. it would better to indicate by color changes in the word file then change into pdf. Some figs. are not visible

English is fine

Author Response

The comments have been addressed in Rebuttal. 
